# Multipolar lasing modes from topological corner states

Ha-Reem Kim[1,4], Min-Soo Hwang [1,4], Daria Smirnova [2,4], Kwang-Yong Jeong[1], Yuri Kivshar [2✉] & Hong-Gyu Park [1,3✉]

Topological photonics provides a fundamental framework for robust manipulation of light, including directional transport and localization with built-in immunity to disorder. Combined with an optical gain, active topological cavities hold special promise for a design of light-emitting devices. Most studies to date have focused on lasing at topological edges of finite systems or domain walls. Recently discovered higher-order topological phases enable strong high-quality confinement of light at the corners. Here, we demonstrate lasing action of corner states in nanophotonic topological structures. We identify several multipole corner modes with distinct emission profiles via hyperspectral imaging and discern signatures of non-Hermitian radiative coupling of leaky topological states. In addition, depending on the pump position in a large-size cavity, we generate selectively lasing from either edge or corner states within the topological bandgap. Our studies provide the direct observation of multipolar lasing and engineered collective resonances in active topological nanostructures.

[1] Department of Physics, Korea University, Seoul 02841, Republic of Korea. [2] Nonlinear Physics Center, Research School of Physics, Australian National University, Canberra, ACT 2601, Australia. [3] KU-KIST Graduate School of Converging Science and Technology, Korea University, Seoul 02841, Republic of Korea. [4] These authors contributed equally: Ha-Reem Kim, Min-Soo Hwang, Daria Smirnova. ✉email: ysk@internode.on.net; hgpark@korea.ac.kr

opological states were originally discovered in condensed matter physics[1,2], but later they have attracted a lot of attention in photonics offering a novel approach in the manipulation of light[3,4]. In particular, the specific photonic structures termed as "photonic topological insulators" (PTIs) support the directional transport of light along interfaces through symmetry-protected edge states that show strong immunity to backscattering from disorder[5–10]. This motivates the use of photonic topological edge states for transmitting optical signals with suppressed losses. A new class of topological materials, higher-order topological insulators, exhibits a dimensional hierarchy of robust topological states, being able to trap localized states at their corners[11–20]. Two approaches have been suggested to construct second-order topological insulators in planar systems. The first approach is based on the quadrupole bulk polarization originally designed by introducing a negative coupling to emulate an artificial magnetic flux[11,14,15,18]. This route however remains challenging to implement in subwavelength photonics. The other approach is related to the dipole polarization quantized by the Wannier centers in distorted lattices[12,13,16,17,19,20]. Direct imaging of such corner states was performed by near-field scanning measurements in second-order topological photonic crystals made of dielectric rods[13,20–22].

Topology-driven localization of optical states unlocks special prospects to facilitate lasing[23] and quantum light generation[24] in active photonic structures with improved reliability. To date, topological active cavities have been probed in different platforms, including polariton micropillars[25], ring resonators[26,27], waveguide arrays[28], and magnetically biased photonic crystals[29], with a focus on lasing from edge states. Nanostructured semiconductor photonic crystals with embedded optical gain are now emerging for multiple applications of nanophotonics[30–32]. The pioneering studies employed one-dimensional Su–Schrieffer–Heeger (SSH) designs to demonstrate light generation driven by mid-gap edge states[30,31]. Most recently, directional vertical emission was devised from a bulk state trapped by a hexagonal topological cavity implemented in a patterned InGaAsP slab[32]. To further reduce the cavity size and improve the laser performance in planar landscapes, here we employ tightly confined corner states with small mode volumes.

Here, we demonstrate lasing action of topological corner states in a rationally designed photonic cavity. Theoretical analysis and numerical simulations predict the existence of four multipole corner modes, which stem from the coupling between the corner states in a square-lattice topological cavity characterized by distinct quantized Zak phases in the interior (nontrivial) and exterior (trivial) domains. In experiment, we consistently observe all of these four corner-state lasers using our hyperspectral imaging system and unambiguously identify them through comparison with the simulation results. In particular, we demonstrate dipole corner-state lasers with a diagonal antinode intensity. Furthermore, both corner-state and edge-state lasing is exhibited in a larger-size photonic cavity.

## Results

**Multipole corner modes**. We base our design on the two-dimensional SSH model[12,17] on a square lattice, in which the unit cell represents a symmetric quadrumer composed of four elements (see Fig. 1a, b). Within the tight-binding (TB) approximation, the topological transition in this lattice is governed by the ratio of the intra-cell $t_b$ to inter-cell $t_a$ couplings (see Fig. 1b). The topological phase can be described by the vector Zak phase or two-component bulk polarization $(P_x, P_y)$: $P_x = P_y = 1/2$ for $t_a > t_b$ (nontrivial), whereas $P_x = P_y = 0$ for $t_a < t_b$ (trivial). A double projection of the polarizations at the corners leads to a corner

charge given by $Q_{xy} = 4P_xP_y$. Therefore, a finite lattice of expanded quadrumers ($t_a > t_b$) with open boundary conditions shows corner states by the nontrivial polarization exactly at zero energy even though they appear embedded in the bulk spectrum[12]. Corner states supported by a corner-shaped domain wall with a weak interface bond can be revealed in the charge probability density of the degenerate zero-energy level (see Supplementary Fig. 1). In the limiting case $t_b = 0$, the system splits into a set of quadrumers in the bulk, whose modes with eigenfrequencies $[-2t_a, 0, 0, 2t_a]$ give rise to the bulk bands, dimers on the edges with energy splitting $[-t_a, t_a]$, and zero-energy monomers on the corners. In real photonic crystals, the chiral symmetry can be broken and edge/corner state frequencies altered by next-nearest-neighbor (NNN) coupling and local boundary effects[12,21,22,33]. To spectrally isolate corner states in the complete bandgap, we assume dimerization $t_a/t_b > 2$ and relatively strong interface tunneling $t_w$ ($t_b < t_w < t_a$), and also incorporate distance-decaying NNN interactions, that yields spectra qualitatively similar to refs. [21,22,33]. Corner states residing in the topological bandgap can be visualized by calculating their profiles on a finite lattice with an embedded square-shaped domain wall (see "Methods" section). In the TB-model analysis, given coupling of the corner states, we obtain four multipole corner modes, namely the quadrupole, two degenerate dipoles, and monopole modes, as shown in Fig. 1c. According to irreducible representations for point group $C_{4v}$, the modal profiles are assigned the eigenvectors $(1, -1, 1, -1)$, $(1, 0, -1, 0)$, $(0, -1, 0, 1)$, and $(1, 1, 1, 1)$, respectively.

In an open electromagnetic system, these multipole localized corner modes can be described by the non-Hermitian coupled-mode equations

$$i\partial_t \Phi = H_{CMT}\Phi = \begin{pmatrix} \omega_c - i\gamma & -t_c & 0 & -t_c \\ -t_c & \omega_c - i\gamma & -t_c & 0 \\ 0 & -t_c & \omega_c - i\gamma & -t_c \\ -t_c & 0 & -t_c & \omega_c - i\gamma \end{pmatrix} \begin{pmatrix} \phi_1 \\ \phi_2 \\ \phi_3 \\ \phi_4 \end{pmatrix},$$
(1)

where the four-component function $\Phi$ comprises the amplitudes of the individual corner states $\phi_i$ (numbering is clockwise). The diagonal elements of $H_{CMT}$ constitute the Hamiltonian for non-interacting corners with $\omega_c$ and $\gamma$ being the resonant frequency and decay rate of the uncoupled corner states, respectively. The off-diagonal non-Hermitian part of $H_{CMT}$ describes interactions with complex coupling strength $t_c = t_r + it_i$, where $t_r$ and $t_i$ quantify spatial mode overlapping and coupling through the radiation continuum, respectively. Once Eq. (1) is solved for the stationary case assuming $t_i, \gamma \ll t_r$, we obtain four eigenstates, which correspond to Fig. 1c, and possess different imaginary parts of eigenfrequencies. In particular, the leakage of the quadrupole mode appears suppressed compared to the uncoupled states.

**Topological cavity design**. Next, we design a topological cavity by imprinting a SSH lattice of air holes in a dielectric slab (Fig. 1d). A topologically nontrivial domain constituted by two types of square air holes with side lengths $(l_1, l_2)$ is surrounded by a trivial one with the inverted order $(l_2, l_1)$, as shown in Fig. 1d. The designed square-shaped domain wall contains 4 corners. We specifically consider a relatively small cavity with the nontrivial domain of $6 \times 6$ unit cells to minimize the number of bound states. Numerical simulations are performed for the InGaAsP slab of finite thickness 275 nm with the following lattice parameters: $a = 500$ nm (lattice constant), $l_1 = 0.58a$, and $l_2 = 0.29a$ (see "Methods" section). The photonic bandgap of the infinite periodic structure opens in a wavelength range around 1550 nm. A

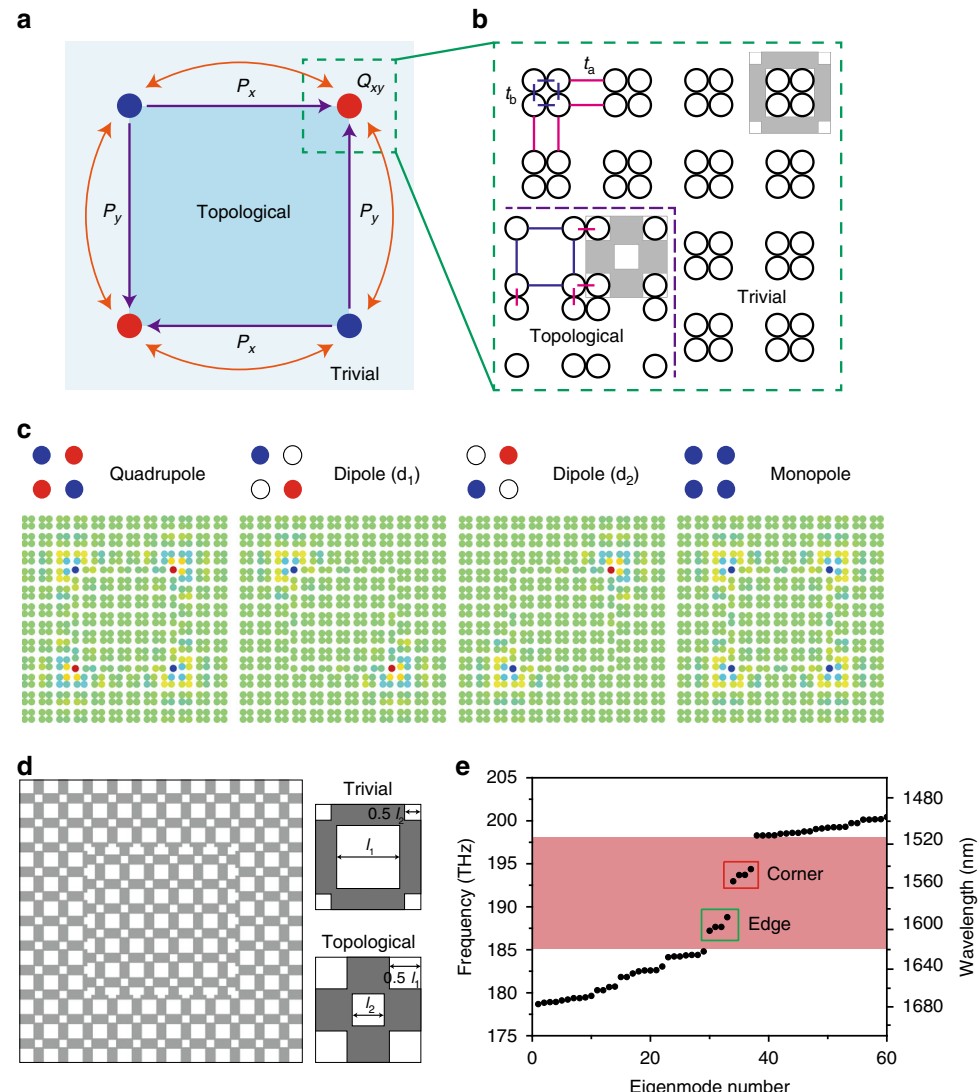

**Fig. 1 Topological cavity design. a** Schematic of a square PTI cavity: a closed topologically nontrivial domain is embedded into the trivial structure. $P_x$ and $P_y$ illustrate edge polarization and $Q_{xy}$ is the corner charge. The orange arrows denote the coupling between neighboring corners. **b** SSH-lattice implementation of **a**: $t_a$ and $t_b$ denote intercellular and intracellular coupling strengths of nearest-neighboring elements, respectively. The purple dashed line indicates the interface between the trivial and nontrivial domains. **c** Visualization of optical corner modes in the structure using the tight-binding model. **d** Schematic of the PTI cavity made of 6 × 6 unit cells in a dielectric membrane (left). The trivial and topological unit cells (right) consist of two types of square air holes with side lengths of ($l_1$, $l_2$) and ($l_2$, $l_1$), respectively. **e** Spectrum of eigenmodes computed for the parameters $a = 500$ nm, $l_1 = 0.58\,a$, $l_2 = 0.29\,a$, and $h = 275$ nm, where $a$ is the lattice constant and $h$ is the slab thickness. The bandgap (magenta-shaded frequency range) hosts 4 corner states and 4 edge states.

topological phase transition between the two lattices is demonstrated by band inversion at the high-symmetry X point of the Brillouin zone (see Supplementary Fig. 2). The complete photonic bandgap hosts interface-bound edge states as shown in supercell simulations (see Supplementary Fig. 3).

Figure 1e shows the calculated discrete spectrum of the modes in the topological cavity. The bandgap structure comprises 4 corner and 4 edge modes within the topological bandgap. The simulated out-of-plane magnetic-field profiles of the topological corner modes (see Supplementary Fig. 4a) are classified by their symmetry in compliance with TB-model analysis (Fig. 1c). The remaining edge modes are confined at the boundaries of the cavity (Supplementary Fig. 4b). The quadrupole corner mode exhibits the highest quality (Q) factor among the corner states. This is a direct signature of radiative coupling between the corners that, as follows from the coupled-mode-theory model,

leads to the suppression of leakage from this eigenstate and should facilitate its lasing when gain is introduced into the structure.

**Photoluminescence measurements for spectral imaging.** To experimentally verify the performance of the designed corner-state lasers, we nanofabricate a topological cavity using an InGaAsP slab incorporating three quantum wells (see "Methods" section). A scanning electron microscope (SEM) image of the sample is shown in Fig. 2a. As designed in Fig. 1d, the cavity consists of the topologically nontrivial structure surrounded by the trivial structure with a lattice constant of 500 nm. The length of the interface between the trivial and topological structures is 3 μm, which is designed to be slightly smaller than the pump spot size of ~3.5 μm. Then, photoluminescence (PL) measurements are performed using a 980-nm pulsed pump laser at room

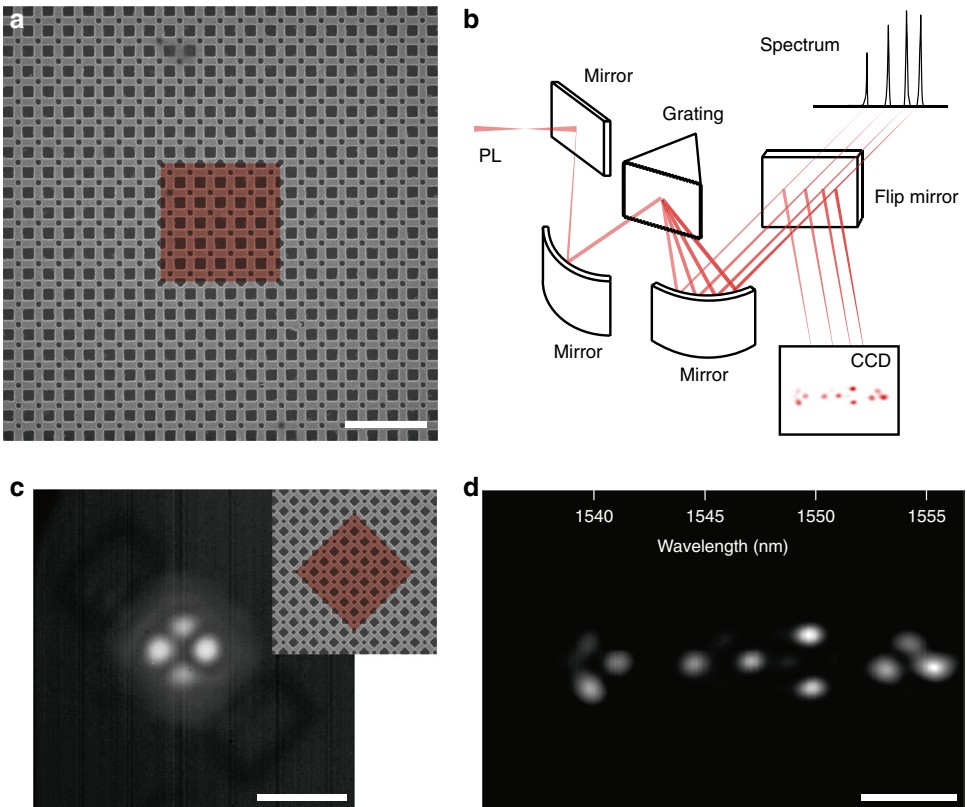

**Fig. 2 Photoluminescence measurements. a** Scanning electron microscope (SEM) image of a fabricated PTI cavity with 6 × 6 topological unit cells. The lattice constant, length of the large square hole, and length of the small square hole are 500 nm, 307 nm, and 158 nm, respectively. The red-color region (false color) indicates the nontrivial domain. Scale bar, 2 μm. **b** Schematic of the spectral imaging measurement setup. **c** Measurement of an emission profile without using the spectral imaging setup. Scale bar, 10 μm. Inset, SEM image of the 45°-rotated PTI cavity, which was used for the optical measurement. The red-color region (false color) indicates the nontrivial domain. **d** Spectral imaging of all excited lasing modes in **c**. Four lasing modes are observed at the wavelengths of 1539.6, 1546.1, 1549.9, and 1553.9 nm. Scale bar, 10 μm.

temperature. The light emitted from the topological cavity is collected by a 50× long-focal objective lens with a numerical aperture of 0.42 (see "Methods" section). To analyze the excited multiple cavity modes individually, we built an optical measurement setup for spectral imaging (Fig. 2b). In this setup, the output signals of the multi-modes are dispersed into spectral components in image space by a spectrometer grating, and the image and spectrum of each mode are recorded at each wavelength with a spectral resolution of <0.6 nm (see "Methods" section).

We examine the topological cavity of Fig. 2a using the spectral imaging system. The cavity is rotated by 45° to more clearly distinguish each individual lasing mode (Fig. 2c, inset). To excite all possible modes simultaneously, the pump laser is focused with sufficient power on the cavity center, including the four boundaries of the topological insulator domain. When the grating is not used (and a mirror is used instead), we observe a conventional mode image in which the images of all the excited modes are superimposed (Fig. 2c). After introducing the grating, the multi-mode images are separated with respect to the wavelength (Fig. 2d). Four lasing modes with different wavelengths appear at different image locations. We note that the spectral imaging system enables the efficient isolation of a single mode without requiring sensitive adjustments of the pumping position.

**Optical properties of corner-state topological lasers**. To identify the four lasing modes in Fig. 2d, we measure the mode image,

lasing spectrum, and light in-light out (L-L) curve of each mode (Fig. 3). We also use the spectral imaging setup to measure the optical properties of each mode separately. Then, we observe the following key features. First, the measured mode images exhibit intensity antinodes confined at 2 or 4 corners (Fig. 3a, d, g, j; left). The 2-corner antinodes exist at the horizontal or vertical diagonal corners (Fig. 3d, g). Second, these lasing modes are all operated in single modes (Fig. 3b, e, h, k). We observe sharp single peaks solely at 1539.6 (b), 1546.1 (e), 1549.9 (h), and 1553.9 nm (k), respectively, although they were originally closely located together in the lasing spectrum. Third, a clear lasing behavior is observed for each mode at lasing thresholds ranging from 400–460 μW (Fig. 3c, f, i, l). The quadrupole mode is experimentally confirmed to have the lowest lasing threshold, due to the increased $Q$ factor originated from the coupling between the corners. Fourth, there is little polarization effect in the quadrupole and monopole modes, whereas the dipole modes exhibit the preferred polarization directions (Fig. 3b, e, h, k; insets).

We compare the measured mode images with the calculated electric-field intensity profiles of the corner-state modes (Fig. 3a, d, g, j; right). The measured images and the corresponding calculated images agree very well. Also, the measured and calculated resonant wavelengths are almost identical. Such good agreement between the measurements and simulations indicates that the measured modes in Fig. 3a, d, g, j correspond to the corner-state quadrupole, dipole ($d_1$), dipole ($d_2$), and monopole modes, respectively. We observe split wavelengths of the degenerate dipole corner-state modes ($d_1$ and $d_2$). The wavelength split originates from the slightly different

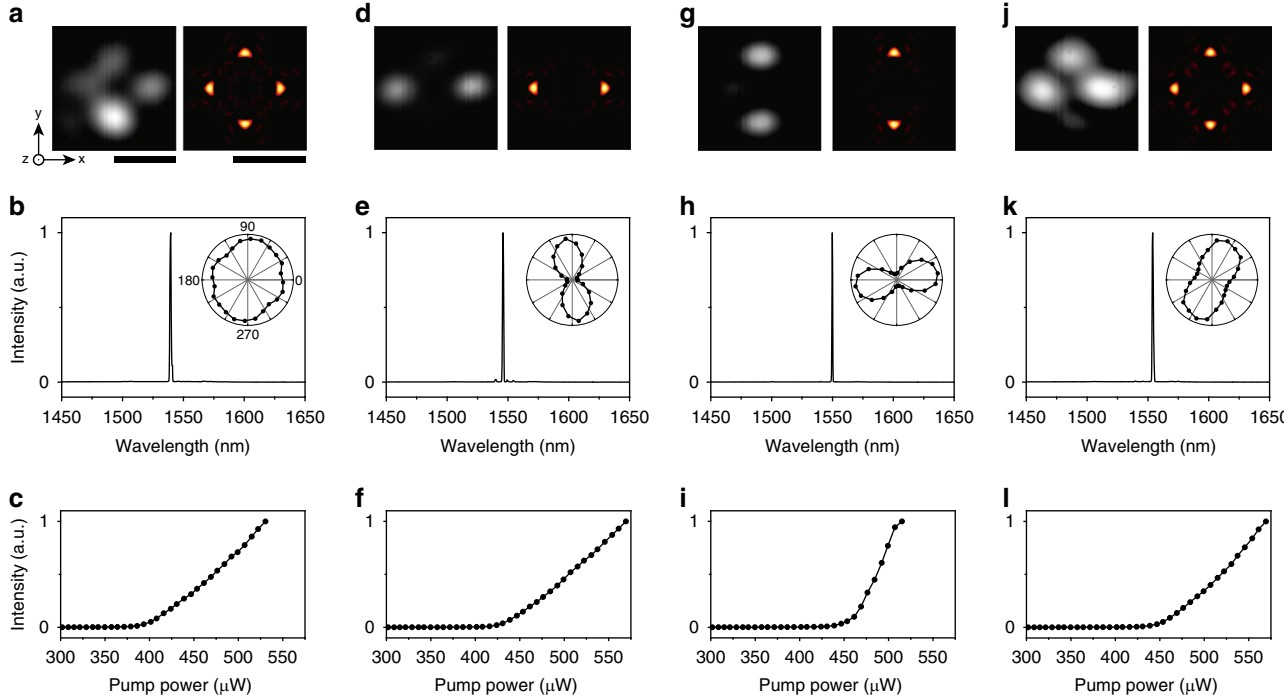

**Fig. 3 Corner-state topological lasers.** Optical properties of the topological quadrupole (**a–c**), dipole ($d_1$) (**d–f**), dipole ($d_2$) (**g–i**), and monopole lasing modes (**j–l**). **a**, **d**, **g**, **j** Measured mode images (left) and calculated z-components of the Poynting vectors (right). The calculation is performed at a z-position of 300 nm above the slab. Scale bars, 5 μm (left and right). **b**, **e**, **h**, **k** Measured above-threshold PL spectra. The peak wavelengths are 1539.6 nm (**b**), 1546.1 nm (**e**), 1549.9 nm (**h**), and 1553.9 nm (**k**). The spectral linewidth was resolution-limited in the spectrometer. Insets, measured output intensity of each mode as a function of the polarization angle using a linear polarizer in front of the spectrometer. 0° and 90° are the horizontal and vertical directions, respectively. **c, f, i, l** Measured L–L curves.

coupling strengths between the corners due to the fabrication imperfection. However, their unique and characteristic mode profiles, such as intensity antinodes at the diagonal corners, are clearly observed in the experiment. It can be thought that the spatial symmetries are only 0.25% broken during fabrication, if we consider the small split wavelength of the degenerate dipole corner-state modes. Consequently, corner-state lasing modes spectrally located close to one another are unambiguously identified by comparing spectral imaging measurements with numerical simulation results.

The rationally designed corner-state lasers show remarkable optical features which differentiate them from edge-state topological lasers[25,26]. First, the corner-state lasers exhibit lower lasing threshold due to smaller mode volume and higher Q factor. For example, the measured threshold in Fig. 3 is more than 4 times and more than 80 times lower than those in ref. [26] and ref. [25], respectively. Second, contrary to the edge states, collective topological modes can be facilely designed by controlling the coupling strength between corners. As a result, here we observe multipole corner-state lasing modes with distinct emission profiles and optical properties. Furthermore, the improved stability to structural imperfections and disorder, being one of the driving forces for topological photonics, is clearly revealed in the corner states operation[14,20,21,34]. We experimentally observe the multipole corner-state lasing modes even when a defect is introduced at the interface between the trivial and nontrivial domains (see Supplementary Fig. 5). In simulations, after introducing a defect even by one unit cell close to the corner, the corner modes remain within the bandgaps with negligible frequency shifts (Supplementary Fig. 6). The corner states are also found well-defined even in the presence of displacements of holes and irregularities in the sizes of holes at the edges, up to 10%

relative perturbations (Supplementary Fig. 7). Besides, being deeply confined, these corner states well sustain distributed bulk disorder (Supplementary Fig. 8). It is not feasible to systematically observe these features in typical photonic-crystal defect lasers.

**Lasing from edge and corner modes in a large-size cavity.** Next, we examine a larger topological cavity with 12 × 12 topological unit cells to investigate both corner-state and edge-state modes (Fig. 4a). Similar structural parameters to those in the topological cavity with 6 × 6 topological unit cells (Fig. 2a) are employed, but the length of one side of the topological domain is 6 μm in this cavity. The PL measurements are then performed by scanning the pump laser along the interface between the trivial and nontrivial structures, because the pump spot size is smaller than the cavity size in this case. When the pump laser illuminated the corners or edges of the interface, bright emission from the cavity is observed at the pumping position (Fig. 4b, c). Four modes are strongly confined at each corner (Fig. 4b), and four other modes are confined at each edge (Fig. 4c). We also measure the PL spectra from these corner-state and edge-state modes (Fig. 4d). Each mode exhibits a single sharp lasing peak: the four corner-state peaks have similar wavelengths of ~1455 nm, whereas the wavelengths of the four edge-state peaks are ~1495 nm. Furthermore, L–L curves with lasing thresholds of 240–290 μW are measured from these modes, showing a clear lasing behavior and transition from spontaneous to stimulated emission (Supplementary Fig. 9).

To further understand the measurements, we perform numerical simulation using finite-element method (Supplementary Fig. 10). We introduce an optical loss in the slab and a local

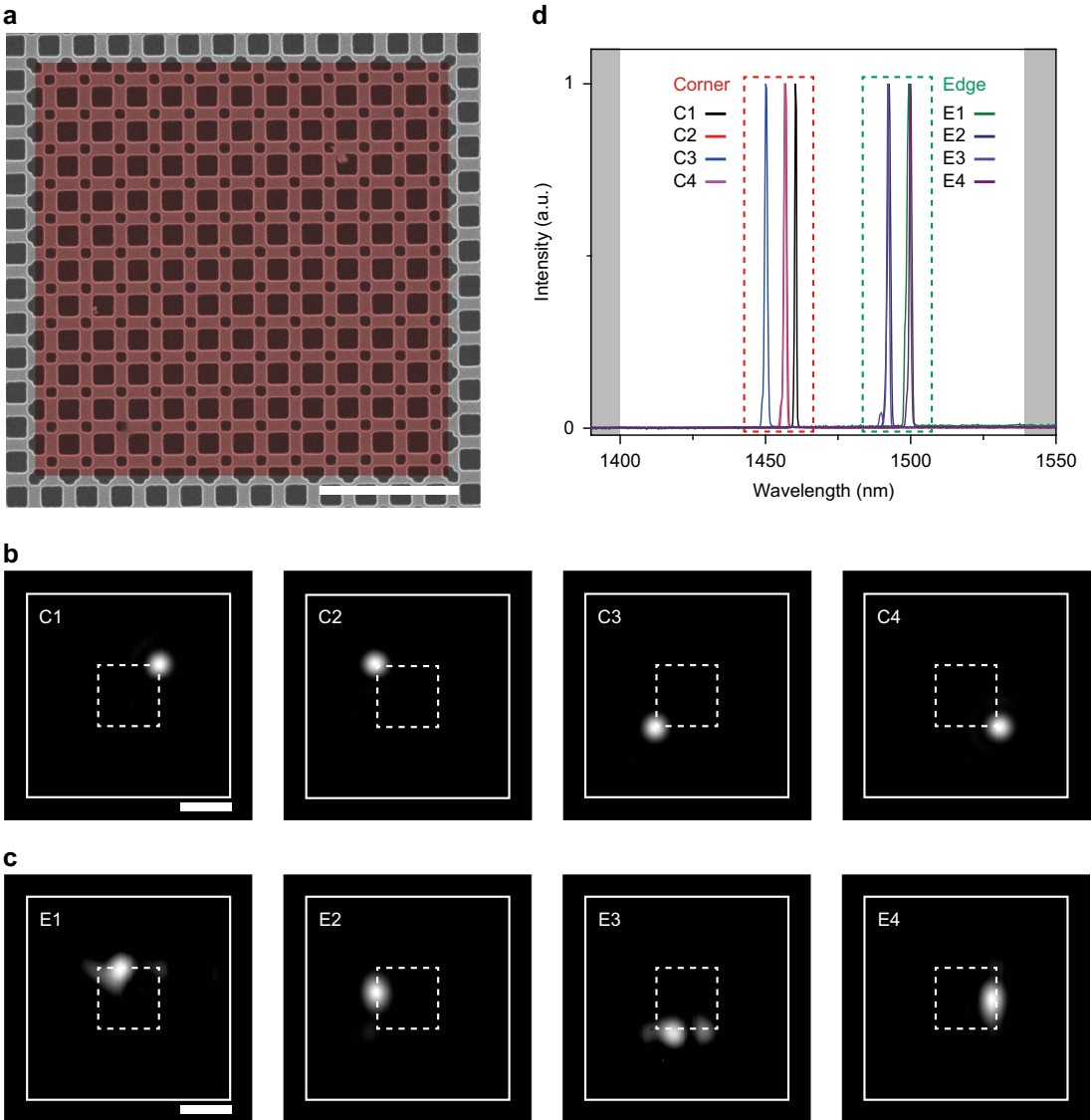

**Fig. 4 Lasing from edge and corner modes. a** SEM image of the fabricated PTI cavity with 12 × 12 topological unit cells. The lattice constant, length of the large square hole, and length of the small square hole are 500 nm, 312 nm, and 158 nm, respectively. The red-color region (false color) indicates the nontrivial domain. Scale bar, 2 μm. **b**, **c** Measured mode images of corner-state lasers (**b**) and edge-state lasers (**c**). The pump laser illuminated the cavity at the positions of the emitted spots. The white dashed lines indicate the interface between the trivial and nontrivial structures, and the white solid lines indicate the outer boundary of the trivial structure. Scale bars, 5 μm. **d** Measured PL spectra of the corner-state and edge-state lasers. The wavelengths of the four corner-state mode peaks and the four edge-state modes peaks are ~1455 nm and ~1495 nm, respectively. The calculated bandgap is denoted by the white region between the gray ones.

gain at the corner or edge to simulate the optical pumping in the PTI cavity with 12 × 12 topological unit cells. Then, the electric-field intensity profiles are calculated depending on the pumping position: the calculated emission profiles at the corner and edge agree well with the measured ones. Also, the calculated wavelength difference between the corner and edge modes is comparable to the measured value. We note that the decreased coupling strength between the corners, due to the optical loss in the unpumped region, yields the field confinement at only one corner, which exhibits the same situation as the experiment.

## Discussion
We have demonstrated the topological corner-state lasers in a square-lattice photonic cavity. Four corner modes formed due to the coupling between the corners have been revealed in the TB-model and full electromagnetic calculations. Lasing actions from these

corner modes have been achieved in the nanofabricated InGaAsP slab with embedded quantum wells. Isolation of the multipole lasing modes has been performed by employing hyperspectral imaging measurements, which enabled the direct mapping and unambiguous identification of the lasing modes. We have also observed light emission from both edge-state and corner-state in a larger-size photonic cavity with reduced coupling between the corners.

Very recently, lasing in a single corner has been reported for similar nanopatterned structures[35,36]. Those corner lasers, however, operate at low temperatures[35] or do not show the unique properties of multipolar lasing in coupled corner modes with different field profiles and Q factors[35,36]. In this work, we have conclusively demonstrated the stable multipole corner modes with antinode intensities at two or four corners as a result of coupling between the corners. In particular, dipole corner-state lasers with a diagonal antinode intensity have been

observed unambiguously. The coupling between the corners has been shown to be controlled and adjusted on purpose by gain/loss contrast distribution over the PTI cavity.

We believe our results open novel prospects for the topologically controlled generation of light in active nanophotonic structures with topological phases and the exploration of radiatively coupled topological states in non-Hermitian systems. Furthermore, combining topological properties and practicability of conventional photonic-crystal defect lasers, corner-state nanocavities can find applications in the implementation of nanoscale lasers with improved reliability, controllability and low lasing threshold.

## Methods

**Numerical modeling.** The tight-binding (TB) calculations (MATLAB) were performed for the two-dimensional SSH lattice of discrete sites connected by alternating weak ($t_b = 1$) and strong ($t_a = 3.3$) bonds. This dimerization yields complete bandgaps in the solvable bulk spectrum $\omega(k_x, k_y) = \pm |t_b + t_a e^{ik_x a}| \pm |t_b + t_a e^{ik_y a}|$, which host gapped edge modes and corner states localized at the open nontrivial boundaries and domain walls. The finite-lattice geometry containing a square-shaped domain wall in Fig. 1a–c consists of two domains created by inversion of the order of strong and weak couplings with additionally incorporated NNN interactions. Importantly, NNN hopping in our model does not violate $C_{4v}$ symmetry of the lattice, and the boundary effects only alter the edge-state and corner-state frequencies (see Supplementary Note 1). The strength of the long-range electromagnetic coupling in real resonant metasurfaces can be tuned by changing the structural parameters. The optical modes and electromagnetic band structures of SSH-like lattices implemented in the perforated InGaAsP membrane were calculated using a three-dimensional finite-element-method (FEM) solver in COMSOL Multiphysics. Floquet periodic boundaries and perfectly matched layers were imposed in the lateral and out-of-plane directions, respectively. The structural parameters were estimated from SEM images of the fabricated samples. The refractive index of the InGaAsP slab was set to the value 3.3.

**Device fabrication.** Photonic topological insulator (PTI) cavities were fabricated on a ~270-nm-thick InGaAsP/1-μm-thick InP/100-nm-thick InGaAs/InP substrate wafer. The InGaAsP slab included three quantum wells with a central emission wavelength of ~1.5 μm. After a 130-nm-thick hydrogen silsesquioxane (HSQ) layer (XR-1541, Dow Corning®) was deposited on the wafer, electron-beam lithography at an electron energy of 30 keV was performed to define a periodic square-lattice structure with hole patterns. The HSQ layer acted as an etch mask for the chemically assisted ion-beam etching, which was used to drill air holes in the InGaAsP layer. The sacrificial InP layer was then selectively wet etched using a diluted HCl:H$_2$O (4:1) solution.

**Optical measurements.** A 980-nm pulsed laser diode (1.6% duty cycle; 1 MHz period) was used to optically pump the fabricated topological cavities at room temperature. The light emitted from the cavities was collected by a 50× long-focal objective lens with a numerical aperture of 0.42 (M Plan Apo NIR B, Mitutoyo) and focused onto a spectrometer (SP 2300i, Princeton Instruments). The grating with 300 grooves mm$^{-1}$ blazed at 1.2 μm was used to spectrally disperse the PL emission from the cavities. The light was sent to either an IR array detector (PyLoN, Princeton Instruments) or an InGaAs IR camera (C10633, Hamamatsu) using a flip mirror in the spectrometer. For conventional mode imaging (not spectral imaging), a mirror was placed instead of the grating.

## Data availability
The data that support the findings of this study are available from the corresponding authors upon request.

## Code availability
The MATLAB codes used in this work are available from the corresponding authors upon request.

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

## Acknowledgements
This work was supported by the National Research Foundation of Korea (NRF) funded by the Korean government (MSIT) (grant 2018R1A3A3000666), and the Australian Research Council (grants DE190100430 and DP200101168).

## Author contributions

Y.K. and H.-G.P. conceived the research and supervised the project. H.-R.K. and M.-S.H. fabricated the samples. H.-R.K. and K.-Y.J. conducted experimental studies. M.-S.H. and D.S. performed numerical simulations and theoretical calculations. M.-S.H., D.S., H.-G.P., and Y.K. wrote the manuscript based on the input from all the authors. All authors contributed to writing and editing the manuscript.

## Competing interests

The authors declare no competing interests.
